# A Novel Family of Lysosomotropic Tetracyclic Compounds for Treating Leukemia

**DOI:** 10.3390/cancers15061912

**Published:** 2023-03-22

**Authors:** José M. Carbó, Josep M. Cornet-Masana, Laia Cuesta-Casanovas, Jennifer Delgado-Martínez, Antònia Banús-Mulet, Lise Clément-Demange, Carme Serra, Juanlo Catena, Amadeu Llebaria, Jordi Esteve, Ruth M. Risueño

**Affiliations:** 1Josep Carreras Leukaemia Research Institute (IJC), 08916 Barcelona, Spain; 2Leukos Biotech, 08021 Barcelona, Spain; 3Faculty of Biosciences, Autonomous University of Barcelona, 08193 Barcelona, Spain; 4Faculty of Pharmacy, University of Barcelona, 08028 Barcelona, Spain; 5MCS, Laboratory of Medicinal Chemistry and Synthesis, Institute of Advanced Chemistry of Catalonia (IQAC-CSIC), 08034 Barcelona, Spain; 6SIMChem, Institute for Advanced Chemistry of Catalonia (IQAC-CSIC), 08034 Barcelona, Spain; 7Department of Hematology, Hospital Clínic, 08036 Barcelona, Spain; 8Institut d’Investigacions Biomèdiques August Pi i Sunyer (IDIBAPS), 08036 Barcelona, Spain

**Keywords:** leukemia, drug discovery, autophagy, apoptosis, new chemical entity, lysosome

## Abstract

**Simple Summary:**

In spite of the recent expansion of the therapeutic landscape for acute myeloid leukemia (AML), resistance mechanisms and relapsed disease still pose a serious barrier to achieve curation for most patients. Considering the high inter- and intrapatient heterogeneity, disruptive therapeutic approaches are expected to provide a clinical solution for this unmet need. An in silico drug discovery program identified a new family of lysosome- and mitochondria-targeting compounds that specifically eradicate leukemia ex vivo and in vivo in relevant preclinical models by inducing both mitochondrial damage and apoptosis, and the simultaneous lysosomal membrane leakiness. Moreover, the compounds are effective in a wide panel of cancer cell lines, as their mechanism of action targets a common neoplastic feature. These compounds possess adequate pharmacological properties rendering them promising drug candidates for AML and unrelated neoplasias, and support their further clinical development.

**Abstract:**

Acute myeloid leukemia (AML) is a heterogeneous hematological cancer characterized by poor prognosis and frequent relapses. Aside from specific mutation-related changes, in AML, the overall function of lysosomes and mitochondria is drastically altered to fulfill the elevated biomass and bioenergetic demands. On the basis of previous results, in silico drug discovery screening was used to identify a new family of lysosome-/mitochondria-targeting compounds. These novel tetracyclic hits, with a cationic amphiphilic structure, specifically eradicate leukemic cells by inducing both mitochondrial damage and apoptosis, and simultaneous lysosomal membrane leakiness. Lysosomal leakiness does not only elicit canonical lysosome-dependent cell death, but also activates the terminal differentiation of AML cells through the Ca^2+^–TFEB–MYC signaling axis. In addition to being an effective monotherapy, its combination with the chemotherapeutic arsenic trioxide (ATO) used in other types of leukemia is highly synergistic in AML cells, widening the therapeutic window of the treatment. Moreover, the compounds are effective in a wide panel of cancer cell lines and possess adequate pharmacological properties rendering them promising drug candidates for the treatment of AML and other neoplasias.

## 1. Introduction

Acute myeloid leukemia (AML) is a very aggressive hematological neoplasm clinically characterized by a high rate of relapses and overall low survival scores. The optimization of chemotherapy regimens and bone-marrow transplantation has improved outcomes during the last few decades, but the recent advent of novel personalized medicine treatments has not brought the promised revolution so far [1,2]. The therapeutic landscape in AML has been greatly expanded the last few years after decades of stagnation, with the regulatory agencies approving eight new therapies, including improvements in chemotherapy, inhibitors of FMS-related receptor tyrosine kinase 3 (FLT3) and isocitrate dehydrogenase (IDH), and drugs targeting BCL-2 and the Hedgehog signaling pathway [3]. While all these therapies were greenlit on the basis of promising preliminary data, most of them have not had the expected clinical success. This is due to limited clinical studies leading to approval (small single-arm studies), the small percentage of patients benefiting from them, and the side effects [4].

In contrast with the very specific mutation-related changes targeted by many novel therapies, there is a growing interest in global alterations in cancer cell organelles and their therapeutic leveraging [5,6]. Leukemic transformation is accompanied by an increase in energetic and biomass demands, leading to major alterations in mitochondrial and lysosomal compartments. Particularly, the expansion of the lysosomal compartment was observed for AML: leukemic cells have larger and more abundant lysosomes that are more sensitive to pharmacological disruption compared to their healthy myeloid counterparts [6]. AML lysosomes also display alterations in their sphingolipid profile [7] and could be implicated in chemoresistance [8]. Regarding leukemic mitochondria, transformation induces an increase in mass and respiration [9,10,11], a reliance on OXPHOS, higher vulnerability to oxidative stress [10,11,12], and changes in the relevant ultrastructural features [13]. Indeed, disrupting the mitochondria (i.e., by inhibiting OXPHOS) was proposed as a therapeutic strategy in AML, and clinical trials are ongoing [11,12,14,15,16], although the appropriateness of this strategy was recently challenged [17].

These unique alterations enable organelle-directed therapeutic strategies, particularly with organellotropic drugs. In this context, we previously identified a group of antihistamines with antileukemic effects grounded on the disruption of lysosomes and mitochondria [14]. Encouraged by those results and conscious of their pharmacological limitations that prevented their indication in leukemia, we aimed to develop novel compounds to unleash the full potential of this new mechanism of action for oncology. Here, we describe the preclinical development of a family of antileukemic tetracyclic compounds with a novel scaffold that display druglike properties while maximizing the antileukemic effects.

## 2. Materials and Methods

### 2.1. Primary Samples

Primary AML samples were obtained at diagnosis from patients diagnosed at Hospital Clínic of Barcelona (Spain) and Hospital Germans Trias i Pujol (Badalona, Spain). AML diagnosis, classification, and risk stratification were based on the latest revised WHO criteria (Appendix A). Clinical characteristics of AML patients are summarized in Appendix A. Mononuclear cells (MNCs) were isolated with Ficoll density gradient centrifugation (GE Healthcare, Chicago, IL, USA). All patients provided their written informed consent in accordance with the Declaration of Helsinki, and the study was approved by the corresponding ethics committees. Mature blood MNCs were isolated from healthy donors’ buffy coats provided by Banc de Sang i Teixits (BSiT, Barcelona, Spain). Umbilical-cord blood was provided by BSiT, and MNCs were depleted for lineage marker-positive cells using magnetic separation with the Human Lineage Cell Depletion kit (Miltenyi Biotec, North Rhine-Westphalia, Germany) following the manufacturer’s recommendations.

### 2.2. Chemoinformatic Screening

The chemoinformatic screening was developed by Chemotargets SL using their proprietary software CLARITY®, version 1.0 (© Chemotargets SL.). Briefly, the software was fed with a set of the antihistamines of interest, which allowed for the establishment of pharmacological descriptors defining the group. Then, using those descriptors, the software searched in an inhouse database of available compounds from combinatorial chemistry repositories, and selected hits with good predicted safety and pharmacokinetic profiles. Hit compounds were purchased from ChemBridge and Ambinter. Greater amounts of EDK-87 and EDK-88 were synthesized at the MCS, IQAC-CSIC (Barcelona, Spain).

### 2.3. Cytotoxicity Assays

First, 2 × 10^5^ (72 h assays in cell lines (Appendix A)), 3 × 10^5^ (48 h assays in cell lines) or 7.5 × 10^5^ (primary AML, healthy donor buffy coats) cells per mL were cultured in 96-well plates, and all drugs were added at the indicated concentrations. In α-tocopherol experiments, cells were treated with growing concentrations of α-tocopherol and EDK drugs for 48 h. In the NAC and 3MA experiments, treatment was simultaneous with EDK compounds. In glucose deprivation experiments, cells were cultivated with a standard medium (glucose high, 2 g/L), a standard medium without glucose, or a standard medium with galactose (2 g/L) and without glucose. Cell viability was measured via 7-AAD (eBioscience) exclusion and Hoechst33342 (Sigma-Aldrich, St. Louis, MO, USA) positivity staining via flow cytometry, and the cell count was obtained through volume. In the experiments with the primary cells, analyses were performed inside the blast population as detected via blast gates (CD45^low^ SSC^int^). For solid tumor cell lines, viability was analyzed using CellTiter-Glo^®^ Luminescent Cell Viability Assay (Promega).

### 2.4. In Vivo Studies

We myeloablated 6–8-week-old NOD.Cg-*Prkdc^scid^ Il2rg^tm1Wjl^*/SzJ (NSG, Jackson Laboratories) mice with busulfan (30 mg/kg IP) at Day 0. The next day, mice were transplanted with KG1 (10 million cells per mouse) or MonoMac1 (1 million per mouse) AML cells that had previously been transduced with luciferase-containing plasmid pLL-EF1a-rFLuc-T2A-GFP (Systems Bioscience #LL410PA-1) and left untreated for 3 (MonoMac1) or 5 days (KG1). Mice were then treated as indicated in the figures. For IP administration, compounds were dissolved in DMSO and diluted in saline prior to injection. For SC treatments, drugs were solubilized using Tween 80 (Escuder) and high-viscosity carboxymethylcellulose (CMC, Merck). For bioluminescence detection, mice were SC-administered 150 mg/kg Xenolight D-luciferin potassium salt luminescent substrate (Perkin Elmer), and the luminescence of isoflurane-anesthetized mice was acquired after 10 min in an IVIS Lumina II imaging device (Perkin Elmer) at adequate acquisition times (2 min for MonoMac-1, 30 s for KG1). The following quantitation and analysis were conducted with Aura Imaging software (Spectral Instruments Imaging). The last day of the experiment, mice were culled, and tibias and femurs were harvested. Engraftment was determined as the percentage of live human CD45-expressing cells in the bone marrow as assessed with flow cytometry. Mice were randomized and blind-coded at the beginning of the experiment, although the groups were not blinded to the in vivo researcher.

### 2.5. Lysosomal Studies

Lysosomal mass was determined using Lysotracker DeepRed (Thermo Fisher, Waltham, MA, USA), and analyzed with cytometry and microscopy, as explained previously [14]. The lysosomal activity assays were performed following the guidance of the study that first introduced the Lysosomal Metriq probe [18]. AML cell lines were lentivirally transduced with the pCW57.1 Lysosomal-METRIQ plasmid (Addgene #135401), and selected with puromycin (2 µg/mL). Cells were treated for 3 consecutive days with 3 µg/mL doxycycline (Sigma Aldrich, St. Louis, MO, USA), incubated with the compounds of interest for 24 h, and analyzed in the flow cytometer. Lysosomal activity was measured as the ratio between red fluorescence (acting as an internal control) and green fluorescence (located in the lysosomes and degraded to different degrees depending on the activity of the lysosomal compartment). For the Galectin 3 puncta assay, AML cell lines were lentivirally transduced with the mAG-Gal3 plasmid (Addgene #62734). Treatments and analysis were performed as described elsewhere [14]. Then, 2 × 10^5^ cells per ml were pulsed with dextran-rhodamine (0.25 mg/mL) for 24 h, chased for 2 h, treated with EDK compounds at the indicated concentrations for 48 h, attached to poly-L-lysine (50 μg/mL)-coated chambered coverslips (μ-Slide 8 well, Ibidi), imaged in a Zeiss LSM880 microscope, and analyzed with Fiji software.

### 2.6. Autophagy Analysis

Protocols for Cyto-ID staining and LC3 detection were previously described [14]. Antibodies used for Western blot are listed on Appendix A.

### 2.7. TFEB Activation

We transfected 2.5 × 10^5^ HEK-293T cells per condition using the jetPEI DNA transfection reagent (Polyplus) with TFEB promoter-luciferase reporter Addgene#66801. Then, at 24 h post-transfection, cells were treated at the indicated conditions, and the 48 h post-transfection luminescence of lysed cells was analyzed according to the manufacturer’s specifications. 

### 2.8. Mitochondrial Analysis

Seahorse studies were performed using the Agilent Seahorse XF Real-Time ATP Rate Assay Kit. Briefly, MonoMac1 and HL60 cells were suspended in complete medium at a concentration of 5 × 10^5^ cells per ml and treated with EDK-87/EDK-88 with the indicated compounds for 18 h. Cells were washed and suspended in Seahorse medium (Seahorse XF medium with 2 mM glutamine, 10 mM glucose, 2 mM pyruvate) at a concentration of 1.4 × 10^6^ (Mono Mac1) and 2.4 × 10^6^ (HL60) cells per mL; 50 μL per well were seeded in 96-well Seahorse plates precoated with polylysine, centrifuged, and left to adhere for 20 min at 37 °C before adding prewarmed Seahorse medium to a final volume of 180 μL per well. Basal OCR and ECAR were measured, and the ATP rate assay was performed with the sequential injection of 1.5 μM oligomycin and 0.5 μM rotenone/antimycin A. A detailed description of this protocol was provided by the manufacturers. Mitochondrial ROS was quantified using the MitoSOX Red Mitochondrial Superoxide Indicator (Thermo Fischer Scientific, Waltham, MA, USA), as previously described [14]. The activation of executive caspases was determined using CaspaseGlo^®^ 3/7 Assay (Promega), following the manufacturer’s recommendations. The detection of Annexin-V staining was also described previously [14].

### 2.9. ADMET and PK Profiles

All ADMET studies were performed in Eurofins Discovery Services (CEREP laboratories, Celle-Lévescault France). The pharmacokinetic studies were performed in Draconis Pharma SL (Barcelona, Spain) in CD-1 mice treated with 30 mg/kg EDK87 or EDK88 via SC or OG. Blood was collected by cardiac puncture at 0.5, 1, 2, 4, or 24 h. 

### 2.10. Synergy Calculation

Synergy was calculated using SynergyFinder 2.0 with the Bliss reference model.

## 3. Results

### 3.1. Identification of Novel Antileukemic Molecules

On the basis of the structures of antihistamines previously identified as antileukemic agents [14], we searched for novel potential antileukemic drugs in catalogues from chemical providers encompassing more than 12 million compounds using CLARITY^®^v1.0 software (Chemotargets SL, Barcelona, Spain). We selected a total of 114 compounds that shared key physicochemical and molecular properties with the input antihistamines, and had a better safety and pharmacological profile based on the in silico prediction. We then screened those compounds in the HL60 (acute myeloblastic leukemia with maturation, FAB M2), KG1 (undifferentiated AML, FAB M0/1) and MonoMac1 (acute monocytic leukemia, FAB M5) AML cell lines as a representation of the heterogenicity in AML. Ten of them belonging to diverse chemical classes, reduced the number of live cells by >50% after 48 h of treatment at 10 µM in at least one cell line (Figure 1A). We selected two of those hits for further study (EDK-87 and EDK-88), given their powerful effects in all AML cell lines and their similar chemical structures. Both had a sulfur-containing tetracyclic hydrophobic head and a hydrophilic tail with a tertiary amine moiety (Figure 1B). On the basis of this common structure, they can be considered cationic amphiphilic drugs (CAD), a label shared with antileukemic antihistamines and the majority of lysosomotropic drugs [14,19]. To further characterize and validate their antileukemic properties, dose–response viability curves were performed on a wider panel of 7 AML cell lines representing all major subtypes of the disease and diverse differentiation status. Viability curves at 48 h revealed EC50 values ranging between 3 and 11 µM, which was effective in all assayed lines (Figure 1C,D), independent of subtype and maturation state. As a further validation of the antileukemic effect of the compounds, EDK-87 (10 and 25 µM) and EDK-88 (5 and 10 μM) were assayed in 14 primary AML samples and 9 peripheral blood mononuclear cells from healthy donors’ buffy coats used as healthy counterparts. Both EDK-87 and EDK-88 greatly reduced the numbers of primary leukemic cells (43% ± 8.5 and 97% ± 1.4, respectively), while healthy-donor blood cells were less affected at some conditions (33.2% ± 8.1 and 55.8% ± 16.7, respectively), demonstrating the existence of a clear and significant therapeutic window (EDK-87: 2.4; EDK-88: 15.8), especially for EDK-88, for which the statistical significance was stronger (Figure 1E and Appendix A). To measure the most primitive population of cells with self-renewal and differentiation capacity, we performed clonogenicity assays with primary AML samples and lineage-depleted umbilical cord blood cells as a source of primitive hematopoietic cells. Again, treatment with the selected compounds was effective in AML (EDK-87: 77.9% ± 9.2; EDK-88: 59.5% ± 19.9), and a therapeutic window was maintained (EDK-87: 3.2; EDK-88: 1.5) (Figure 1F).

### 3.2. In Vivo Effectivity of EDK-87/EDK-88 

To faithfully determine the relevance of antileukemic therapies, it is essential to test them in their physiological cellular context, accounting for the key influence of microenvironmental factors. Therefore, once the antileukemic effects had been validated ex vivo, we studied their effectivity in vivo in adult conditioned immunodeficient NSG mice transplanted with rLuc-transduced KG1 and MonoMac1 AML cell lines. The KG1 model is a cytarabine-sensitive slow-growing AML, and cytarabine-resistant MonoMac1 cells generate a very aggressive disease. EDK-87 and EDK-88 were administered at 30 mg/kg either intraperitoneally or subcutaneously, and the leukemic burden was assessed with both bioluminescence and bone-marrow engraftment, which provide a general idea of the evolution of the disease through time and an endpoint snapshot of the situation in the physiological environment, respectively. As shown in Figure 2, the EDK-87 and EDK-88 compounds effectively reduced the disease progression in mice xenotransplanted with either AML model, and significantly reduced the leukemic burden in the bone marrow (Figure 2A,B and Appendix A), regardless of the chemoresistance and in vivo proliferation rate (EDK-87: 58.44%; EDK-88: 43.06%). The engraftment growth was disrupted in the chemoresistant MonoMac1 AML model (Figure 2A) compared to the deceleration observed in the chemosensitive KG1 model (Figure 2B); since lysosomes have recently been implicated in chemoresistance processes [20], and because previously described lysosomotropic antihistamines [14] were used as the scaffold for the drug design program, the lysosomal compartment was studied in response to resistance acquisition via long-term cytarabine exposure. As suggested from the efficiency assays, a significant increase in lysosomal mass was observed upon the acquisition of cytarabine resistance (Appendix A).

### 3.3. Dual Targeting of Lysosomes and Mitochondria

Since these new compounds were identified on the basis of previous lysosome-affecting drugs [14], we then explored EDK-mediated effects on lysosomes. First, the lysosomal compartment was studied as a whole using a Lysotracker probe, and, as is expected for lysosomotropic drugs, an expansion of the lysosomal mass was observed following treatment with EDK-87 and EDK-88 (Figure 3A,B and Appendix A). This was further validated via an increase in lysosomal activity detected with the lentivirus-encoded Lysosomal-METRIQ probe (Figure 3C and Appendix A), a multicistronic biosensor containing lysosomal protein LAMP1 fused to sfGFP, a T2A self-cleaving peptide and mCherry. Upon transduction, mCherry was released into the cytosol (red internal control), and LAMP1-sfGFP was translocated to the lysosomes (green signal). An increased relative red/green fluorescence ratio indicates the augmented trafficking and degrading activity of lysosomes. We also observed the activation of autophagy, as assessed via LC3-II immunoblotting, a marker of mature autophagosomes, and CytoID, a live probe that stains all autophagic compartments (Appendix A). Nonetheless, conventional autophagy inhibitor 3-MA was not capable of reverting the cytotoxic effects of the EDK compounds (Appendix A), indicating that, while the autophagic compartment was expanded, it was not the main culprit for programmed cell death. These results suggest an accumulation of our compounds in the lysosomal compartment, as anticipated due to its CAD properties. However, lysosome-dependent antitumoral effects are determined via lysosomal membrane permeabilization (LMP) rather than mere lysosomal expansion [21]. As such, LMP was assessed with golden-standard methods: galectin-3 puncta assay and dextran release [22]. In both types of experiments, we detected the clear dose-dependent disruption of lysosomes, manifested through an increase in the formation of galectin-3 puncta (Figure 3D,E and Appendix A) and a change in the distribution of dextran from a punctate into a diffuse pattern (Figure 3F and Appendix A), which is consistent with a leakage of lysosomal contents into the cytoplasm. To further decipher the role of LMP related to antileukemic potential, we combined EDK-87 and EDK-88 with α-tocopherol, which promotes lysosomal membrane stability. α-Tocopherol almost completely abrogated the cytotoxic EDK-87/EDK-88 effects (Figure 3G and Appendix A). Therefore, our novel compounds caused cell death at least partially by permeabilizing lysosomal membrane, sharing the mechanism of action with their predecessors and highlighting the importance of lysosomes in the pharmacological effect of EDK compounds.

Aside from inducing cell death through massive LMP, several lysosomotropic drugs engage in more finetuned signaling mechanisms, namely, releasing calcium to the cytoplasm [23] and activating the master transcriptional regulator of lysosomal biogenesis, TFEB [24]. These two mechanisms are also thought to be mechanistically linked: calcium activates calcineurin, which in turns activates TFEB [25]. Interestingly, when we interrogated these two mechanisms, we detected both a fast increase in intracellular calcium (Figure 3H and Appendix A) and the activation of TFEB activity (Figure 3I) upon EDK-87 and EDK-88 treatment. EDK-87 induced few changes in calcium release, in concordance with its lower potency. Interestingly, chloroquine, a well-described lysosomal disruptor, failed to mobilize intracellular calcium, suggesting a different mechanism of action.

Recently, TFEB has emerged as an important regulator of AML differentiation through its action on MYC. Indeed, TFEB and MYC negatively regulate each other in a tightly coordinated circuit in which increased TFEB represses MYC to induce differentiation and cell death [26]. In fact, inducing the terminal differentiation of AML induces AML cell death [27], constituting the basis of differentiation therapies [28]. To assess whether EDK compounds were modulating this axis, the transcriptional levels of TFEB and MYC were analyzed at different treatment time points. Concordant with the proposed mechanism, EDK-87 and EDK-88 first concurrently induced an increase in TFEB expression with a decrease in MYC (Figure 3J and Appendix A). To check the effects on differentiation, we analyzed the surface expression of myeloid differentiation marker CD11b after 72 h of EDK treatment. Consistent with the proposed model, CD11b was also increased (Figure 3K and Appendix A), and terminal differentiation was further confirmed via morphological analysis using May–Grünwald–Giemsa staining (Appendix A). In addition to morphological changes associated with myeloid differentiation (increased cell size, lower nucleus/cytoplasm ratio), the staining revealed prominent cytoplasmatic vacuolization that was compatible with the previously detected lysosomal expansion (Figure 3A,B and Appendix A). Altogether, these results suggest that EDK compounds trigger differentiation and cell death through a Ca^2+^–TFEB–MYC axis, as depicted in Figure 3L.

Once the effect on the lysosomes had been validated, we studied mitochondria as the second arm of the hypothesized dual mechanism of action. First, we measured mitochondrial respiratory activity with Seahorse and observed that, after treating the AML cell lines with EDK-87 and EDK-88, there was a drastic reduction in oxygen consumption (OCR) (Figure 4A and Appendix A), indicating an impairment of the respiratory chain. This result was accompanied by an increase in mitochondrial ROS (Figure 4B and Appendix A), a consequence of deficient electron transport in the inner mitochondrial membrane, and the activation of apoptosis, as shown by the increase in annexin-V positive cells and the consequent activation of effector caspases 3/7 (Figure 4C–E and Appendix A). Moreover, the cotreatment of EDK-87 and EDK-88 with mitochondrial ROS scavenger *N*-Acetyl-cysteine (NAC) partially reverted the cytotoxicity of these compounds (Figure 4F and Appendix A), suggesting that mitochondrial disruption and consequent ROS generation are necessary for the induced cytotoxicity upon EDK treatment. 

Targeting mitochondria and specifically inhibiting the respiratory chain were proposed as a therapeutic strategy on the basis of the fact that AML cells are generally OXPHOS-dependent [16,29]. To further analyze the relevance of the observed decrease in OCR, we examined the bioenergetic effect of different carbohydrates, enhancing AML cell reliance on OXPHOS by culturing them in glucose-deprived galactose-containing media, switching the main energetic source from glucose into galactose [30]. In those conditions, AML cells were generally more sensitive to EDK compounds, particularly in HL60 (Figure 4G and Appendix A), pointing to the importance of OXPHOS inhibition on the antileukemic properties of EDK compounds. Altogether, these results suggest that EDK-87 and EDK-88 sabotage mitochondria and induce apoptosis, validating their dual mechanism of action that combines apoptotic cell death triggered by the mitochondria and lysosome-dependent cell death.

### 3.4. Synergism with Arsenic Trioxide

Synergistic combinations of drugs allow for the potential widening of the therapeutic window, an increase in effectivity, and a decrease in administered doses [31]. To find drugs that could synergize with our compounds in a real clinical setting, we searched for therapeutics that are approved for hematological malignancies, and target lysosomes and/or mitochondria. Among them, we focused on arsenic trioxide (ATO), a widely used chemotherapeutic agent for treating acute promyelocytic leukemia (APL), since it activates TFEB [32], increases lysosomal activity [33] and the accumulation of acidic vesicles [34], and inhibits OXPHOS [35]. Interestingly, both EDK87 and EDK88 were highly synergistic with ATO in the AML cell lines (Appendix A), and in three primary AML samples while sparing healthy-donor blood cells (Figure 5A and Appendix A). Synergism Bliss scores ranged between 20 and 40, and were detected in a wide range of concentrations for EDK-87, EDK-88, and ATO. Comparing healthy and AML cells, we identified a wide synergic therapeutic window of the combination for EDK-87 (Appendix A) and even more so for EDK-88 (Figure 5B), with a therapeutic index almost reaching 10 in some conditions (meaning that EDK-88 was 10 times more effective in AML than it was in healthy samples). The most interesting results were probably those when treating the samples with 10 µM EDK-87/EDK-88 and 2 µM ATO (Figure 5C and Appendix A). While EDK-88 or ATO alone decreased the viability of AML cells to 58.3% ± 1.7 or 86.9% ± 10.5, respectively, the combination reduced it to 22.43% ± 5.4. In the same conditions, healthy cells remained virtually unaffected. The synergistic effect was also seen in terms of clonogenicity in the AML cell lines (Appendix A). The EDK and ATO combination, therefore, widened the therapeutic window of monotherapies. At the mechanistic level, ATO cooperated with EDKs to increase the lysosomal mass (Figure 5D) and impair OXPHOS, consequently lowering OCR (Figure 5E and Appendix A), which is consistent with previous studies [33,35] and provides a rationale for the synergistic effects. 

### 3.5. Pan-Cancer Effectivity of EDK-87/EDK-88

Considering the previous results showing that our compounds target general tumor vulnerabilities, we speculated that EDK-87 and EDK-88 could possess broad cytotoxic properties against cancer cells. We assayed the cytotoxic effects of EDK-87 and EDK-88 on a panel of cell lines representing several families of solid tumors and non-AML hematological malignancies. We also included 2 nontumor cell lines to account for healthy adherent cells. Both EDK87 and EDK88 affected cell viability in all tested lines (Figure 6A). They displayed greater cytotoxic activity than that of the well-known lysosomotropic chloroquine, and were much milder for nontumor cell lines, further validating the cancer-specific effects and the existence of a therapeutic window (Figure 6B). Moreover, these results suggest that the disruption of mitochondria was necessary for the highly antineoplastic effect of the two compounds compared to a mere lysosomotropic drug, highlighting the existence of a new mechanism of action. Indeed, chloroquine similarly induced cytotoxicity regardless of the neoplastic nature of the treated cells (Figure 6B), possibly explaining the lack of clinical benefit in cancer patients [36]. The case of EDK-88 is particularly notorious, since it eliminates >60% of neoplastic cells in all tested tumor types while greatly sparing fibroblast-like HEK-293T and nontransformed bone-marrow stromal HS-5 cells. EDK-88 is especially effective in B- and T-cell malignancies. 

### 3.6. Pharmacokinetic and ADMET Profiles

To establish the feasibility of further translational efforts, we next studied relevant ADME-Tox and pharmacokinetic (PK) parameters. As expected, both compounds were highly hydrophobic, and solubility in PBS was, in concordance, markedly low, although the results were better for the more realistic gastric and intestinal fluids (Appendix A). Both compounds had modest permeability and intermediate clearance, which was later validated in the PK studies. Regarding hERG inhibition, concerning results were observed for EDK-87, while EDK-88 inhibition levels were within a reasonable margin for an antitumoral therapy (Appendix A). None of the compounds violated Lipinski’s rule of five (Appendix A). The PK profile was determined in mice after administering 30 mg/kg of either EDK-87 or EDK-88 orally and subcutaneously. EDK-87 showed a high volume of distribution and high clearance, resulting in an overall low plasma level and thereby less exposure of the cells to the compound (Appendix A). In contrast, while EDK-88 was characterized by poor oral pharmacokinetics, subcutaneous administration led to higher levels of compounds in the plasma that were completely cleared at 24 h (Appendix A). These differences might account for the higher metabolic rate and lower solubility for EDK-87 as compared to those of EDK-88. Although both compounds are eligible for further preclinical development, EDK-88 constitutes a more desirable and druglike compound.

## 4. Discussion

In spite of the recent expansion of the therapeutic landscape for AML, resistance mechanisms and relapsed disease still pose serious barriers to achieving curation for most patients. Indeed, considering the pervasive intratumor heterogeneity in AML, the wide repertoire of pathways underlying malignancy, and the protective effects of the tumor microenvironment, novel targeted therapies are confronted with escape mechanisms that limit their long-term efficacy [37,38,39,40]. Here, we identified a novel family of effective antileukemic compounds that simultaneously target lysosomes and mitochondria, thus exploiting differential vulnerabilities in AML related to subcellular-level alterations. These innovative compounds, which also activate differentiation through TFEB, are highly synergistic with arsenic trioxide, and possess interesting pharmacological properties that render them promising drug candidates for leukemia and other tumor types.

A limitation of many targeted therapies is their restricted application, as only subsets of patients with specific molecular features are eligible. Additionally, the appearance of chemoresistance-related mutations that abrogate their efficacy is a major challenge using this type of approach. EDK-87 and EDK-88 target organelles that are altered in the majority of AML cases regardless of their differentiation or mutation status [6,29] as a consequence of general transformation events rather than specific molecular changes. Thus, the acquisition of treatment-related resistance and the development of scape mechanisms become highly unlikely. In concordance, EDK compounds were effective in a diverse group of cancer cell lines and primary patient samples representing major disease subtypes. Additionally, the observed therapeutic window when comparing the results to healthy samples further validated the druggability of previously described cancer-specific organelle alterations [6,14,29]. Eradicating the most primitive fraction of cells is of utter importance in order to achieve durable remissions [41,42]. In clonogenic assays, EDK compounds were able to eliminate not only bulk AML blasts, but also the leukemic stem cell subpopulation [14]. More importantly, we provided a solid in vivo proof of concept of the efficacy of the EDK family in two different AML models corresponding to two AML subtypes, including the most aggressive and chemoresistant MonoMac1, suggesting that EDK compounds might be an effective line of treatment for relapse/refractory AML patients, who constitute the main unmet need in this disease [43].

Mechanistically, EDK compounds induce cell death by simultaneously triggering mitochondrial apoptosis and lysosomal membrane permeabilization, validating the previously described mechanism of action [14]. At the signaling level, lysosomotropic drugs induce the lysosomal calcium-release-dependent activation of TFEB [24,25]. Recently, TFEB activation has been described as a key promoter of differentiation and cell death in AML via its modulation of MYC [26]. Here, EDK-87 and EDK-88 triggered a calcium-mediated signaling pathway that led to terminal differentiation through the modulation of TFEB activation and the subsequent downregulation of MYC, following the recently proposed role of lysosomes as key intracellular signaling hubs [44]. MYC, is not only able to induce leukemogenesis [45], but is also responsible for sustaining OXPHOS in AML stem cells [46] that are critical for their survival. Additionally, differentiation therapies are still a promising strategy for inducing leukemic cells to overcome their block in differentiation and become chemosensitive, and activating programmed cell death.

Antihistamines, the drugs originally identified as lysosomal/mitochondrial-targeting agents, have a suboptimal pharmacological profile for a potential reposition to leukemia [14]. Moreover, their antileukemic properties were associated to the prodrug chemical structure and not the actual drug [14]. The newly developed family of EDK compounds substantially improve the efficacy and therapeutic window of their predecessors, overcoming their major limitations, and, more importantly, eliciting their effects in vivo. Preliminary ADMET and PK profiles demonstrated sufficient plasma levels and favorable drug properties, particularly via subcutaneous administration, in contrast to the original lysosomotropic antihistamines [47]. Our work supports the further development of the EDK chemical space, particularly EDK-88 in combination with ATO. In order to optimize these treatments, future studies will improve the pharmacological parameters by testing different counterions that impact pharmacokinetics [48] and evaluating the encapsulated synergic ratios of the compounds. Regulatory preclinical research to advance to a Phase I clinical trial for AML is envisaged in the near future.

## 5. Conclusions

A new chemical space was identified that specifically and efficiently eradicated leukemic cells by targeting lysosomes and mitochondria. EDK-87/EDK-88 presented compatible ADME-Tox and a PK profiles with further clinical development, and a wide therapeutic window.

## Figures and Tables

**Figure 1 cancers-15-01912-f001:**
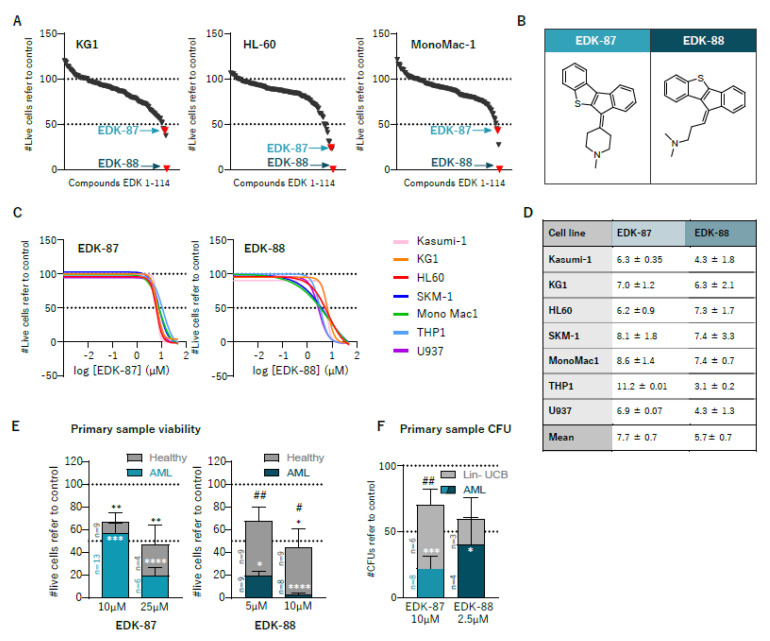
Identification of two hit compounds with antileukemic activity. (**A**) KG1, HL60, and MonoMac1 cells were treated with each of the 114 selected compounds in silico at 10 µM for 48 h, and cell viability was assessed with flow cytometry. (**B**) Chemical structure of EDK–87 and EDK–88. (**C**) AML cell lines treated with increasing concentrations of EDK–87 and EDK–88; cell viability was assessed, and (**D**) EC50 (µM) was calculated. (**E**) AML patient and healthy donor samples were treated with EDK–87 and EDK–88 at the indicated concentrations for 72 h, and cell viability or (**F**) clonogenicity was determined. Lin–UCB: lineage–depleted umbilical cord blood. * *p* < 0.05; ** *p* < 0.01; *** *p* < 0.001; **** *p* < 0.0001 in Kruskal–Wallis tests comparing to untreated control. # *p* < 0.05; ## *p* < 0.01 in Kruskal–Wallis tests comparing healthy and AML.

**Figure 2 cancers-15-01912-f002:**
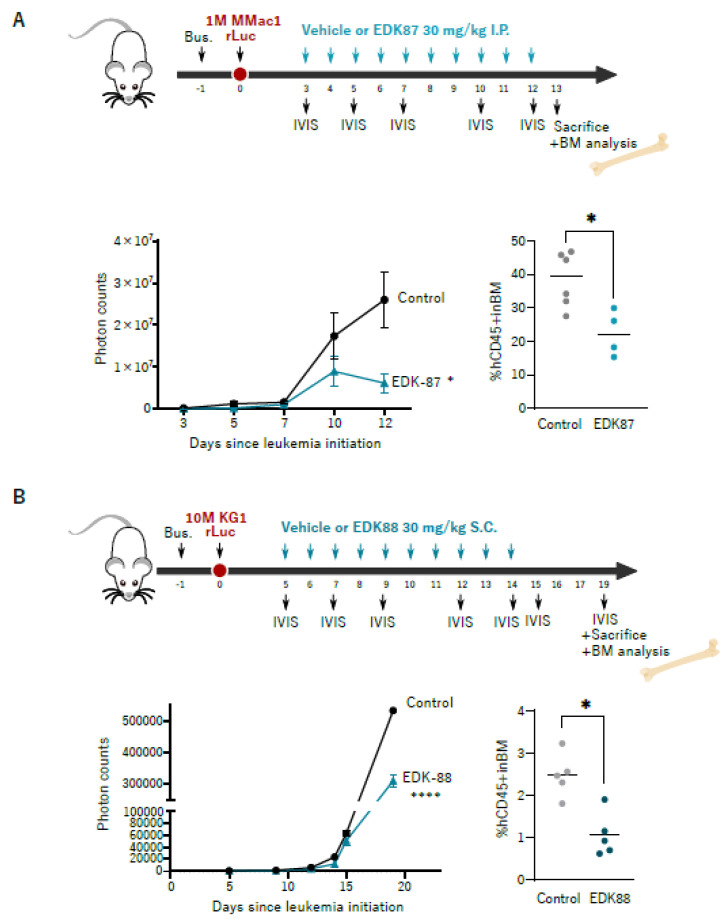
EDK–87 and EDK–88 are effective in vivo. (**A**) Adult conditioned NSG mice (4 mice/group) transplanted with 1 × 10^6^ Rluc–transduced MonoMac1 cells, left untreated for 3 days, and then IP administered with 30 mg/kg EDK–87 for 10 consecutive days. Luminescence was analyzed at Days 3, 5, 7, 10, and 12, and photon counts are represented. On Day 12, mice were euthanized, and the engraftment in the bone marrow was assessed via flow cytometry. (**B**) Adult conditioned NSG mice (5 mice/group) transplanted with 10 × 10^6^ rLuc–transduced KG1 cells, left untreated for 5 days, and then SC administered with 30 mg/kg EDK–88 for 10 consecutive days. Luminescence was analyzed at Days 5, 9, 12, 14, 16, and 19, and photon counts refer to luminescence at Day 5. On Day 19, mice were euthanized, and leukemic engraftment in the bone marrow was assessed with flow cytometry. Bus: busulfan. BM: bone marrow. Luminescence statistics: * *p* < 0.05, **** *p* < 0.0001, in two-way ANOVA. Flow cytometry statistics: * *p* < 0.05 unpaired *t*-test.

**Figure 3 cancers-15-01912-f003:**
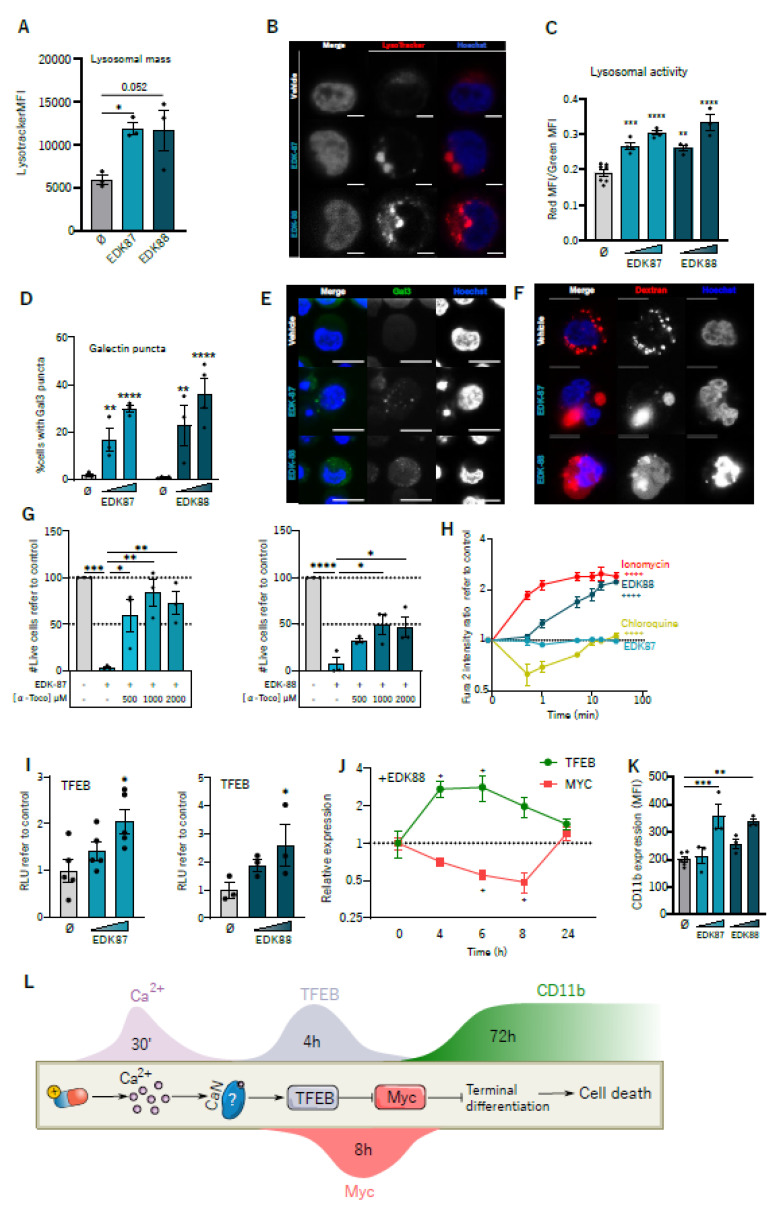
EDK–87 and EDK–88 induce lysosomal–dependent cell death. HL60 cells were treated with EDK–87 or EDK–88 at 10 µM for 24 h, stained with Lysotracker Green, and analyzed with (**A**) flow cytometry and (**B**) confocal microscopy (scale bar = 10 µm). (**C**) LysosomalMetriq–transduced HL60 cells were treated with the compounds at 5 and 10 μM for 24 h, and red/green fluorescence was measured with flow cytometry. (**D**) Gal3–GFP–transduced HL60 cells were treated with the indicated compounds at 10 and 20 µM for 24 h, and cells with galectin 3 puncta were counted via confocal microscopy. (**E**) Representative galectin 3 puncta images (scale bar = 10 µm). (**F**) HL60 cells loaded with dextran–rhodamine B were treated with EDK compounds at 10 µM for 24 h and analyzed via confocal microscopy (scale bar = 10 µm). (**G**) HL60 cells were cotreated with α–tocopherol (α–toco) at the indicated concentrations in the presence of 10 µM EDK–87 (left) or EDK–88 (right) for 48 h, and cell viability was assessed with flow cytometry. (**H**) HL60 cells were loaded with a Fura2 probe and treated with 10 µM EDK–87/EDK–88, 50 µM chloroquine (negative control), or 1 µM ionomycin (positive control). Fluorescence was recorded at the indicated time points. The ratio (340/508)/(380/508) is represented. (**I**) HEK293T cells were transiently transfected with a TFEB reporter vector, treated with 10 and 20 µM EDK–87 and EDK–88 for 24 h, and luminescence was measured. (**J**) HL60 cells were treated at the indicated times (h) with 10 µM EDK–88, and *TFEB* (green) and *MYC* (red) gene expression was assessed with real-time qPCR and normalized to time 0. (**K**) HL60 cells were treated with EDK–87/EDK–88 at 2.5 and 5 µM for 72 h, and surface CD11b expression was analyzed with flow cytometry. (**L**) Proposed mechanism for EDK–induced differentiation. * *p* < 0.05; ** *p* < 0.01; *** *p* < 0.001; **** *p* < 0.0001 in one–way ANOVA except Gal3 puncta (two–way ANOVA) and calcium mobilization (mixed–effects two–way ANOVA with Geisser greenhouse correction).

**Figure 4 cancers-15-01912-f004:**
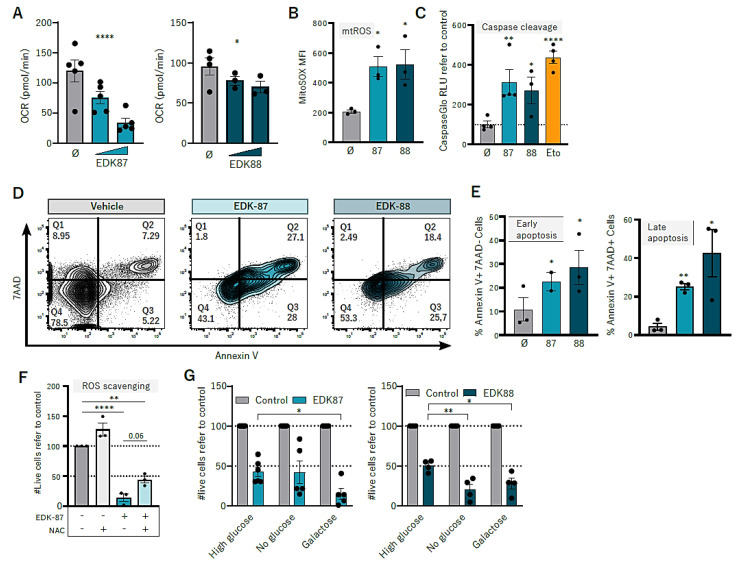
EDK compounds induce mitochondrial damage and apoptosis. (**A**) HL60 cells were treated for 24 h with EDK–87 or EDK–88 at 5 and 10 µM, and basal oxygen consumption rate (OCR) was analyze using Seahorse. (**B**) HL60 cells were treated for 24 h with EDK–87 or –88 at 10 µM, stained with MitoSOX, and acquired in a flow cytometer. (**C**) HL60 cells were treated with 10 µM EDK–87, EDK–88, or etoposide (positive control) for 24 h, and effector caspase activation was measured with CaspaseGlo. HL60 cells were treated for 24 h with EDK–87 or –88 at 10 µM, and apoptotic cells were detected via flow cytometry according to 7AAD and annexin–V staining. (**D**) Representative flow cytometry plots and (**E**) frequency of apoptotic cells. (**F**) HL60 cells were treated for 48 h with EDK–87 10 µM alone or in combination with the ROS scavenger NAC, and viability was assessed via flow cytometry. (**G**) HL60 cells were treated with 10 µM EDK–87 or EDK–88 for 48 h in high–glucose–, no–glucose, or galactose–containing media, and viability was assessed with flow cytometry. * *p* < 0.05; ** *p* < 0.01; **** *p* < 0.0001 in one-way ANOVA except glucose depletion (two–way ANOVA); mtROS: mitochondrial ROS.

**Figure 5 cancers-15-01912-f005:**
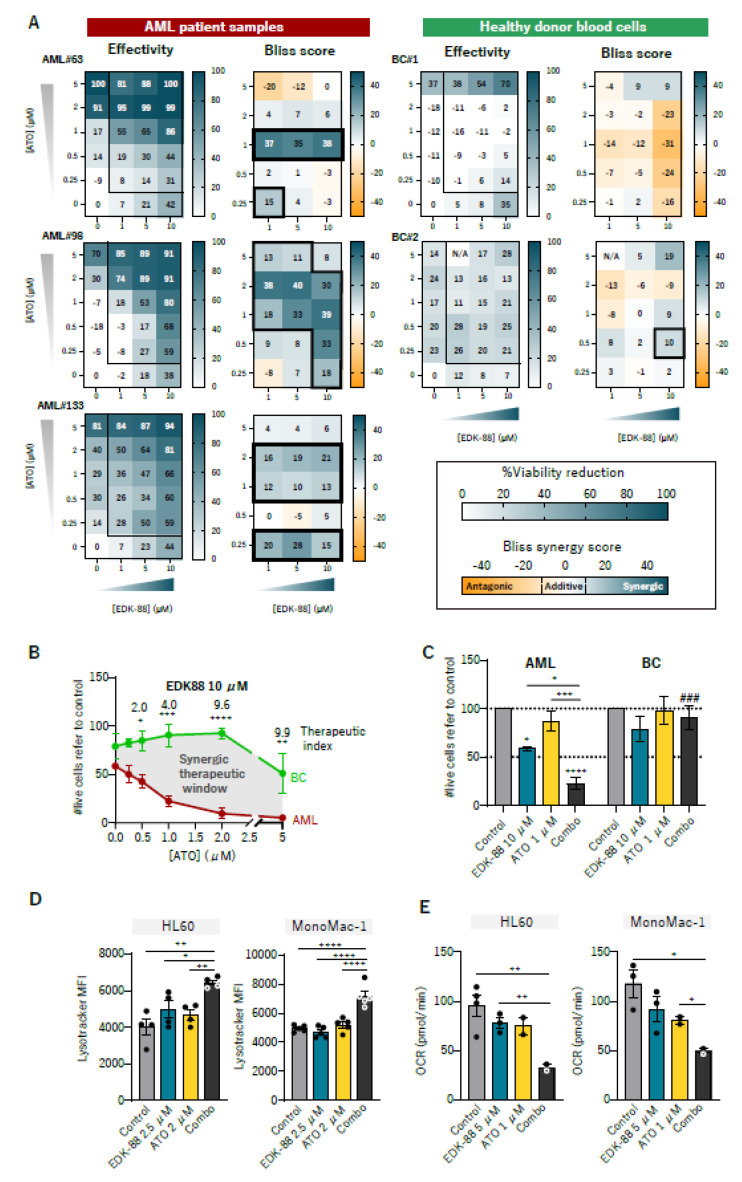
EDK–88 synergizes with ATO in AML cells but not in healthy samples. (**A**) AML patient samples (*n* = 3, left) and two healthy donor blood samples (*n* = 2, right, BC: buffy coat) were treated with combinations of EDK–88 and ATO; viability was analyzed after 48 h with flow cytometry (shown as the frequency of death cells), and synergistic values were calculated (Bliss scores). (**B**) Cytotoxicity effect of the combination of 10 μM EDK–88 and different concentrations of ATO in AML (red) vs. healthy (green) samples in order to visualize the synergic therapeutic window and the therapeutic index (viability of healthy/AML). (**C**) Cytotoxic effects of 10 µM EDK–88, 1 µM ATO, or a combination (combo) in AML samples and healthy blood cells (BC). (**D**) HL60 and MonoMac1 cells were treated for 24 h with 2.5 µM EDK–88, 2 µM ATO, or a combination (combo), stained with Lysotracker, and acquired in a flow cytometer. The mean fluorescence intensity of Lysotracker is represented. (**E**) HL60 and MonoMac1 cells were treated for 24 h with 5 µM EDK–88, 1 µM ATO, or a combination (combo). OCR was measured with the Seahorse technique. * *p* < 0.05; ** *p* < 0.01; *** *p* < 0.001; **** *p* < 0.0001 in (B) two–way ANOVA comparing BC and AML, (C) one-way ANOVA comparing inside families, or (**D**) Welch *t*–test (**E**) ### *p* < 0.001 in two–way ANOVA comparing BC vs. AML.

**Figure 6 cancers-15-01912-f006:**
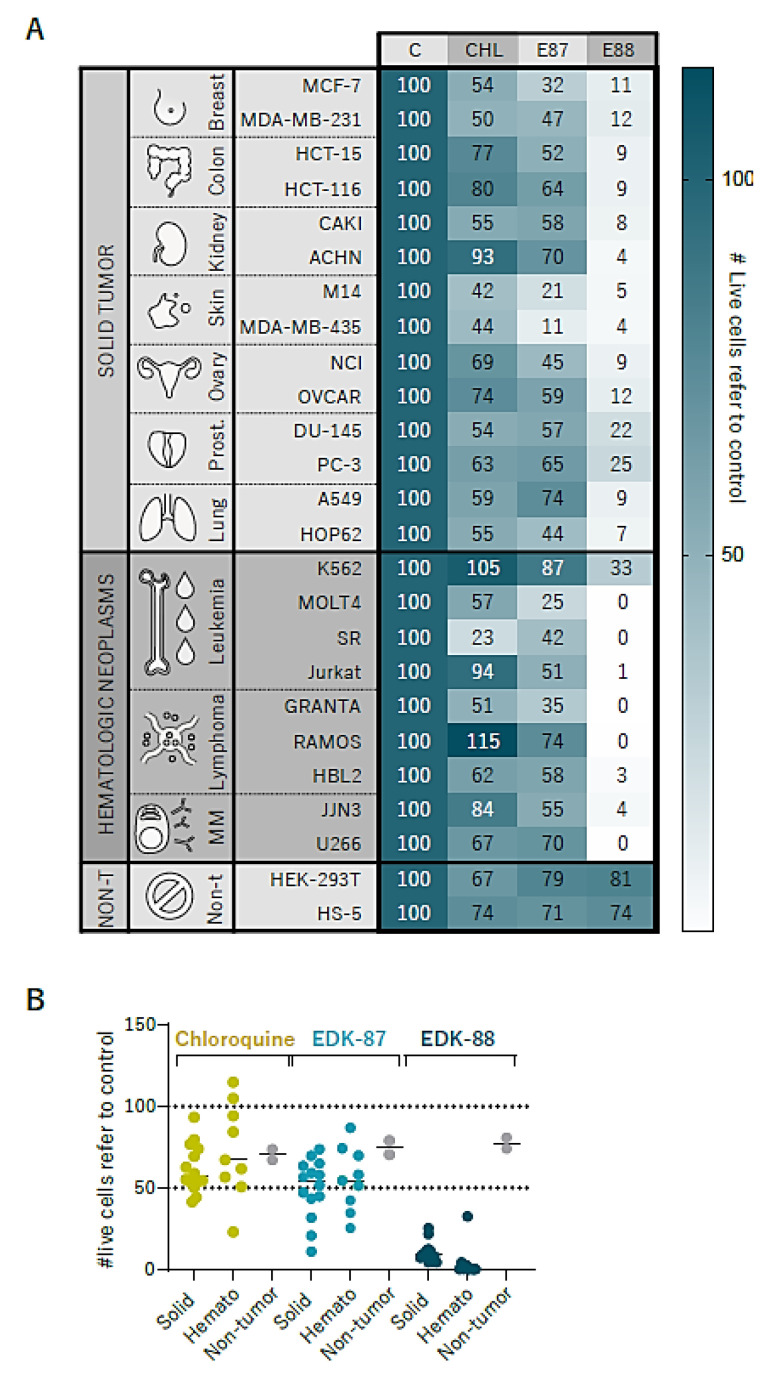
EDK compounds are effective in a pancancer cell line panel. Cancer cell lines were treated with vehicle control (C), 10 µM chloroquine (CHL), 10 µM EDK–87 (E87), or 10 µM EDK–88 (E88), and viability was assessed after 48 h. Experiments were performed in three independent biological replicates; (**A**) means of 3 experiments. Frequency of live cells normalized to control across cell types. (**B**) Previous results grouped by solid tumors, hematological neoplasms, and nontumor cell lines. Each point represents the total mean of a cell line.

## Data Availability

For original data or detailed protocols, please contact risueno@carrerasresearch.org.

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
