# Peer review of "A Novel Family of Lysosomotropic Tetracyclic Compounds for Treating Leukemia"

_cancers, 2023, doi:10.3390/cancers15061912_

Round 1

Reviewer 1 Report

In this article, the authors characterized the antileukemic effects of novel compounds. They focused on the two most efficient compounds (EDK-87 and EDK-88) that reduced cell viability of acute myeloblastic leukemia (cell lines and primary AML samples). The work is interesting and well done.

The authors demonstrated that the compounds target lysosomes and mitochondria. They induced lysosomal membrane permeabilization, the release of calcium to the cytoplasm, the activation of the transcription factor TFEB and the repression of MYC, and the terminal cell differentiation. Whereas the authors clearly demonstrated that EDK-87/EDK-88-induced cell death is dependent on the disruption of lysosomes, they did not demonstrate that cell death is dependent on calcium release and cell differentiation as shown is Figure 3L. Would it be possible that calcium-TFEB-MYC axis-mediated cell differentiation and cell death are independent events, independently induced by lysosome disruption (or by mitochondria disruption)?. To comfort their hypothesis schematized in Figure 3L, the authors could demonstrate that cell death is dependent on calcium release or TFEB activation. The calcium release could be linked to lysosome disruption or mitochondrial disruption or both. Does chloroquine or NAC inhibit calcium release or TFEB activation?

Minors:

·         I cannot find cell line description and culture in materials and methods.

·         Supplementary Figure 2B & C. the legend figure does not correspond to the shown panels.

·         Figure 3 B, E and F: the labelling (x and y) are blurry

·         Supplementary Figure 3D: what is Clq?

·         Supplementary Figure 3F: what is Chl?

·         Supplementary Figure 2C and supplementary Figure 3E are the same.

·         Supplementary Figure legends dos not correspond to the shown graphs: Supplementary Figure 2B, C, 3E, 3F, 3G and 3H.

·         Line 274: it is probably Supplementary Figure 3H instead of 3G.

·         Line 401: Is that 1 or 2µM ATO? Figure 5C indicates 1 µM ATO.

Reviewer 2 Report

The article“A novel family of lysosomotropic tetracyclic compounds for 2 treating leukemia” by José M. Carbó et al. highlights the potential of 2 novel compounds in treating AML and other neoplasias. The compounds are also effective in a wide range of cancer cell lines, and they possess adequate pharmacological properties, which are essential for drug development. The authors also demonstrated these 2 compounds eradicate leukemic cells  through mitochondrial damage and apoptosis, as well as lysosomal membrane leakiness. The dual mechanism of action results in the eradication of leukemic cells through both canonical lysosomal-dependent cell death and activation of the terminal differentiation of AML cells. The article is structured well, and the work is relevant for the readership of ‘Cancers.’ 

There are some questions regarding their description and figures.

1. Figure 1E, EDK-87 reduced the numbers of primary leukemic cells by 43%±8.5 and healthy-donor blood cells were 33.2%±8.1. It’s not a significant difference. The author overstate that it’s a significant therapeutic window. 

2.Figure1F, why did they use 2.5 uM EDK-88 but not 10 uM? As 10 uM is more effective for AML cells as shown in Figure 1E.

3.Did the author test the dose of EDK-87 that has no effect on healthy-donor blood cells but has effect on primary leukemic cells?  Using this dose, how many primary leukemic cells could be eliminated? 

4.For Figure 3J,  did the author test the protein level of TFEB and MYC at indicated times?

5.For the mechanism, EDK compounds induce cell death by triggering mitochondrial apoptosis and lysosomal membrane permeabilization. It’s demonstrated in figure 3 and 4 with in vitro evidence using HL60 cells. Is there any evidence in vivo using leukemic cells from xenotransplanted mice models?

There are some minor mistake listed blow,

Supplementary Figure 2 and 3 are confusing, as figure legends are not matched with figures. There are no figure legends for Supplementary Figure 2C and 3H.

Figure 4E, there are 2 samples for the early apoptosis of 87, while 3 samples for late apoptosis of 87.

Figure 4D, the representative flow cytometry plots of EDK-87 and 88 do not represent the results in Figure4 E as both early and late apoptosis of 88 are higher than that of 87. 
